# Influence of the Strand Characteristics on the Properties of Oriented Strand Boards Obtained from Resinous and Broad-Leaved Fast-Growing Species

**Aurel Lunguleasa** [1,*] **, Adela-Eliza Dumitrascu** [2] **, Cosmin Spirchez** [1] **and Valentina-Doina Ciobanu** [3]

1   Department of Wood Processing and Design of Wood Products, Faculty of Furniture Design and Wood Engineering, Transilvania University of Brasov, 1 Universitatii 1, 500068 Brasov, Romania; cosmin.spirchez@unitbv.ro
2   Department of Manufacturing Engineering, Transilvania University of Brasov, 5 Mihai Viteazul, 500174 Brasov, Romania; dumitrascu_a@unitbv.ro
3   Department of Silviculture and Forest Engineering, Transilvania University of Brasov, 1 Sirul Beethoven, 500123 Brasov, Romania; ciobanudv@unitbv.ro
*   Correspondence: lunga@unitbv.ro

**Abstract:** The paper aims to investigate the influence of the characteristics of the strands on the properties of oriented strand boards (OSB). To solve this objective, some global synthetic characteristics of the strands (the slenderness ratio, the characteristics of thinness, and the specific surface) of four wood species currently used in this technology (spruce and pine for softwood, and poplar and willow for hardwood) were first studied. The characteristics of the OSB obtained from each species separately were also determined, and finally the correlations analysis was made between the characteristics of the strands and those of the corresponding OSB boards. The working methodology used the European tests regarding the physical and mechanical properties of the boards, but also algorithms for forecasting and evaluating the quality of the strands and boards. The conclusions regarding the characteristics of the strands have highlighted the role of the specific surface of the strand and the characteristics of the thickness; respectively, the conclusions regarding the characteristics of the boards showed that the fast-growing species of willow and poplar lead to obtaining higher quality OSB boards. The general conclusion of the paper is that the characteristics of the strands have a significant influence on the physical–mechanical properties of the OSB board.

**Keywords:** strands; oriented strands board (OSB); modulus of rupture (MOR); modulus of elasticity (MOE); internal bond (IB)

## 1. Introduction

Oriented strand board (OSB) are known as 'Sterling board' in the UK and 'Aspenite' in the US and Canada, and they represent tri-layered boards with high mechanical properties used for construction (shacks, temporary construction, sheepfold, false floors, etc.) and are made of long strands of about 80–120 mm. The starting point for their achievement was the wafer-board and flake-board, boards obtained from wide chips with large gluing surfaces, at which the length and orientation of the strands and tri-stratification was performed. The surface of these boards is rough, reason why it is used less often in furniture production. The superior properties and the low price of these boards make it possible to replace the plywood in constructions, even under conditions of high humidity (there are OSB boards for exterior and interior use, according to EN 300 [1]).

The United Nations Economic Commission for Europe (UNECE) [2] showed that the level of plywood and OSB in 2015 was higher in North America than in Europe and CSI, but the production of OSB and MDF (medium-density fiberboard) was much higher in Europe than those two comparison regions (Figure 1). Also, if we compare the production capacities for the countries of Europe and Asia (United Nations Economic Commission for Europe

area), it can see that on the top there are countries like Germany, Romania, and Russia, and countries like France and UK are only in positions 8 and 10, respectively (Figure 2).

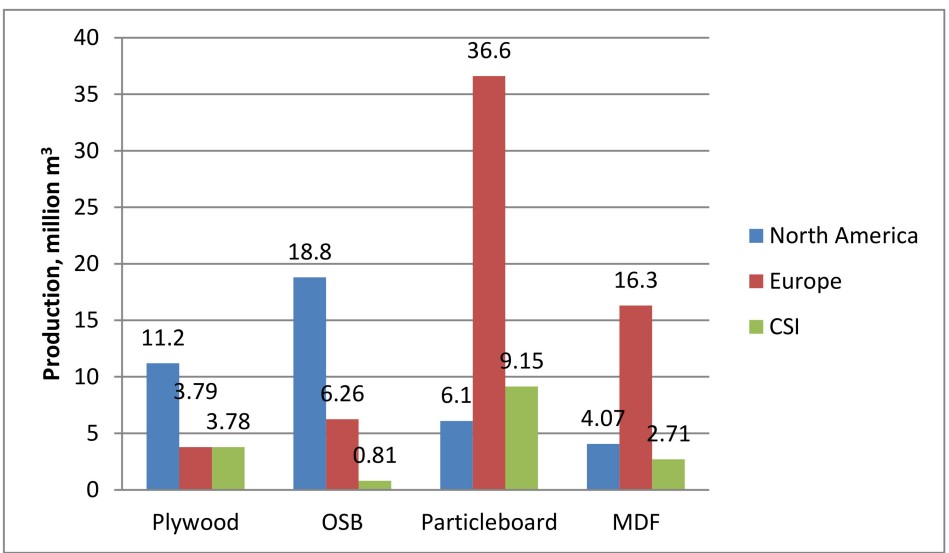

**Figure 1.** OSB production compared to plywood, particleboard, and MDF in UNECE region [3].

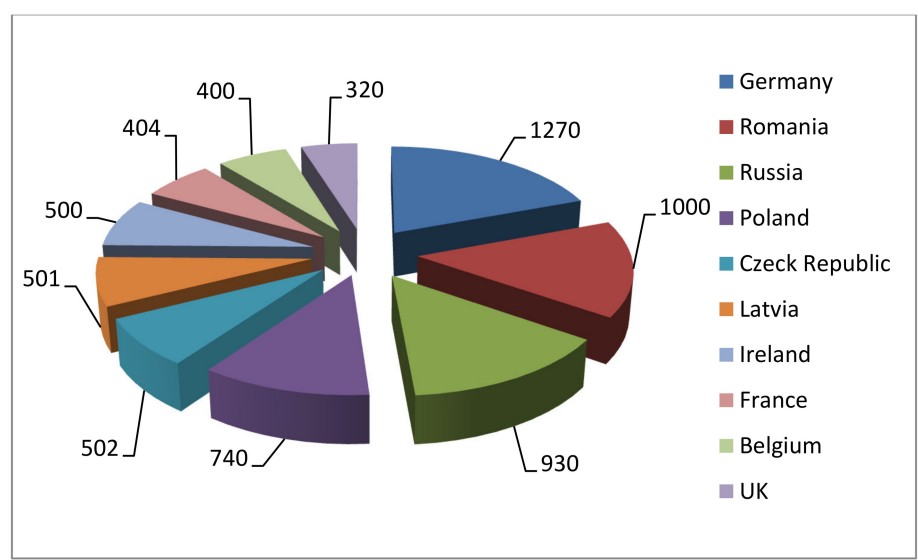

**Figure 2.** Top 10 countries referring to OSB capacities in UNECE zone [3,4].

Some authors [5] studied the influence of the thickness of the strands on the properties of the OSB boards in the lab from cut veneers, finding that the properties of the boards with 0.38 mm thickness of strands were similar to those of the solid wood from which the veneers were cut. The authors did not take into account the fact that in industrial technology the strands are irregular at the ends and thickness, they are aciform, dimensionally varied, and very rough, characteristics that have a positive effect on compaction and densification, obtaining properties much better than in the case of strands obtained from veneers. Other authors [6] analysed the influence of 80 mm long strands of *Pinus taeda* L. on the properties of tri-stratified OSB with the ratio of 1:2 on layers and the density of 0.75 g/cm$^2$. It was found that all mechanical properties exceed the requirements of the standards in the field. Another study [7] showed that the influence of the fineness of the strands observing that the resistance to internal bond (IB), the modulus of rupture (MOR), and the modulus of elasticity (MOE) increased to a level of fineness of 20%. It is also shown that the fine strands have a beneficial influence on the mechanical properties of the OSB boards by

interposing them between the large strands, increasing the cohesion of the layers and compaction of the boards. In reference [8] the influence of three different strand lengths on the properties (MOR and MOE) of OSB boards was examined. The obtained results showed that the obtained properties met the minimum requirements of standard [1] and the length of the strand influenced these properties positively and significantly. Other authors [9] examined the influence of three strands lengths of 78, 105, and 142 mm and two thicknesses of 0.55 and 0.75 mm to observe the influence of strand geometry on the properties of OSB boards. Their conclusion is that a correlation must be made between the long length of the strand that offers good bending resistance and the small length of the strand that offers good compressive strength. The research [10] established that the heat treatment of OSB boards had a positive effect on thickness swelling and equilibrium moisture content (EMC). It has also been shown that immersion in water after 24 h still creates swelling and dimensional changes up to 168 h. Reference [11] studied the methods of eliminating the relaxation of OSB boards after pressing, by the adhesive used type, by increasing the pressing time and by increasing the pressure. The heat transfer mechanisms and the optimum pressing parameters and a linear correlation equation between them were found. Other researchers [12] expressed the area of the strands in $m^2/kg$. The range of variation of the surface of the strands is large, respectively between 15–279 $m^2/kg$ for variations of the thickness of the strands between 0.1–0.9 mm. Theoretical densities of the species were between 300–700 $kg/m^3$. The roughness of the strands was also taken into account. Reference [13] analyzed the use of fast-growing species in obtaining wood-based composites. They also used the technique of heat treatment of wood in order to improve the hygroscopicity properties of wood-based composite. Other authors [14] made a comparison between different categories of composite boards, emphasizing OSB as being cheap, made from wood of poorer quality, and successfully replacing plywood.

Reference [15] studied the influence of heat treatment at 140–180 °C on OSB boards. Reference [16] studied the forest management practice on OSB properties. The control, thin, and fertilized trees were used and it was found that the anatomical structure is different but the mechanical properties are not significantly influenced by the type of wood used. Young et al. [17] studied the increased durability of OSB with White Cedar water extract as a natural fungicide. Reference [18] statistically studied the thickness of the OSB front layer for six plants in the eastern USA. It was found that the mean and median value of the front strand was 0.81 mm, respectively 0.36 mm. Other authors [19] studied the replacement of the core layer of the OSB composition with wood waste or other unsorted particles. The obtained results showed that even large proportions of 75% of these strands in the core structure of the board have led to obtaining OSB boards with good mechanical properties that meet the minimum requirements of standard [1]. Reference [20] analysed the influence of geometry, orientation, and content of thin strands on the properties of OSB boards (MOR, MOE, shear strength technique of simple linear regressions, and that of multi-linear regressions). The obtained results showed that the concentrated static loading of the samples offers conclusive data for interpreting the properties of the OSB boards. Paper [21] analysed the effect of methylene diphenyl isocyanate-MDI (3, 4, 5%, on dry mass) adhesive content on the heat-treated OSB board after fabrication at temperatures of 160 °C and 175 °C. Salari et al. [22] made some evaluations on the use of the Paulownia (*Paulownia fortunei*) wood to OSB production but also on the use of nano-clay particles near UF resin. The obtain results showed a positive effect of nano-clay participation up to 5% of the amount of adhesive. Other research [23] sought to find the optimal orientation of the flakes in the OSB boards in order to find the optimum field of use and it was found that the boards with optimum orientation of the flakes are superior to the commercial ones. Reference [24] studied the introduction of thin strands in the composition of OSB boards for obtaining high performance boards. It has been shown that thin strands have a minimal effect on MOR and MOE, and the steam injection did not have spectacular effects on the properties of the obtained plates. Other research [25] used an image analysis technique to find an OSB board model with a higher quality. Five strand models evaluated

according to their geometry were used and it was demonstrated that there is a correlation between the geometry of the strands, the degree of orientation of the strands in the board and the mechanical properties of the board. Reference [26] analysed the properties of OSB board obtained from a mixture of two European species, namely beech and poplar. The general conclusion is that with increasing the content of beech strands will increase the mechanical properties of the board, but the properties of hygroscopicity will decrease. Reference [27] studied the influence of Poisson's ratios on the characteristics of OSB boards and demonstrated that this method can become a criterion for assessing the quality of these boards.

Reference [28] investigated the effect of the type of adhesive on the combustion of OSB plates based on the ASTM E69 standard and using thermo-gravimetric analysis (TGA). Based on the results of the research, it was found that the lowest temperature was obtained for the phenol-formaldehyde adhesive. Reference [29] studied the effect of the type of adhesive used in OSB technology for the face and core of the plate, noting that the formaldehyde content was limited to that of solid wood. Other research [30] analyzed several physical–mechanical properties of commercial boards measuring 18 mm thick, for the front and core layers, obtaining values over 40% higher in density and more than 2 times the strengths. These results confirm once again that the front layer of the OSB boards differs considerably from the core layer and is the decisive layer for increasing the strengths. Paper [31] studied the influence of density and content of resin content on the mechanical properties of the boards obtained from fast-growing willow (*Salix viminalis*). The effect of density on MOR and MOE was better, while the resin content effect was better only for IB.

The main characteristics of the strands are the geometric ones (length, width, and thickness, expressed in mm), the specific surface of the strands expressed in $m^2/100$ g, the slenderness ratio, and the characteristic of thinness [32]. The main element that gives high resistance to OSB boards is the geometry of the strand used and especially its length, usually around 120 mm. If the thickness of the strands is similar to that used in the classic boards manufacturing from woodchips on average 0.6–0.8 mm, the width of the strands is non-uniform due to their breaking according to the longitudinal plane of minimal resistance of the wood. However, due to the large length of the strands and the use of high-performance cutter with the crown, the strands obtained in OSB technology have a rather large width of 10–12 mm. The slenderness ratio has always been defined as the dimensionless ratio of the length and thickness of the strands and expressed the quality of the strands. In this sense, two strands that have different dimensions but the same slenderness ratio will have the same quality. For example, if it has two strands of different sizes (the first strand with $l_1 = 80$ mm and $t_1 = 0.4$ mm and the second strand with $l_1 = 40$ mm and $t_2 = 0.2$ mm), these strands will have the same slenderness ratio $\lambda = 200$, so they will be of the same quality. Its average values range from 60–120 for the chips used in chipboard technology, from 100–160 for the boards with wide strands and over 160 for the strands of OSB [33]. The thinness characteristic emerged from the desire to take into account the width of the strands (which is also very important in OSB board technology) and was determined as a ratio between the length of the strands and a diameter equivalent to the thickness and width of the strands (Figure 3). The equivalent diameter was determined from the relation of equality of the surface of the circle and of the rectangle.

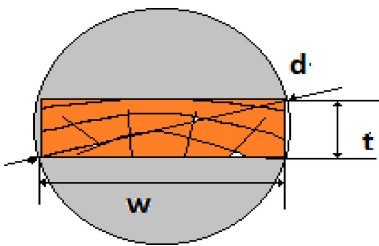

**Figure 3.** Equivalent diameter of the strands.

On the basis of the equivalent diameter, it was found the relationship of determination of the slenderness characteristic (Equation (1))

$$k_t \;=\; l \div 2\sqrt{b \cdot g / \pi} \quad [\mathrm{mm/mm}]$$

(1)

where $k_t$—slenderness characteristic, in mm/mm; $d_e$—the equivalent diameter of the cross-section of the strand, in mm; $l$—the length of the strand, in mm; $g$—strand thickness, in mm; $b$—strand width, in mm.

It was also found, according to a theoretical method, the specific surface of two wood species, used in wafer-board technology. For example, for a thickness of 0.8 mm it was found that the beech strands (*Fagus sylvatica*) have an area of 0.39 m$^2$/100 g and those of poplar will have a specific surface of 0.56 m$^2$/100 g, respectively; it was also observed that once as the density of the wood species decreases, the specific surface of the chips will increase.

Following the bibliographic studies, it can be concluded that, although there are several papers that study the influence of geometric characteristics of strand on the strength of OSB boards, there are no conclusive studies on the influence of global characteristics of chips. Therefore, the main objective of the paper is to find out the influence of the global/synthesis characteristics of some softwood (pine and spruce) and hardwood (poplar and willow) strands on the properties of the OSB boards. From this general purpose, other objectives have been divided, such as finding some global characteristics of the strands (the slenderness ratio, the characteristic of thinness and the specific surface of the strands), as well as their influence on the density, resistance, and elasticity modulus (MOR and MOE) and resistance to internal bond (IB) of OSB boards.

## 2. Materials and Methods

The strands needed for the experiments were taken from Kronospan Trading SRL (Brasov, Romania), immediately after their cutting (Figures 4 and 5), of the four wood species thought to be used: spruce (*Picea abies*), pine (*Pinus sylvestris*), poplar (*Populus tremula*), and willow (*Salix alba*).

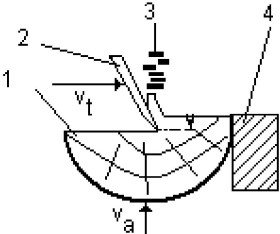

**Figure 4.** Principle of wood chipping: $v_t$—cutting speed; $v_a$—feeding speed; 1—wood material; 2—knife; 3—chip; 4—counter-knife.

These strands were dried up to 10% after which their characteristics were determined taking 100 g of each category. This quantity was divided into four equivalent dimensional quarters and the dimensional characteristics (length, width, and thickness) were determined, and on the basis of these, three main characteristics were determined; namely, the slenderness ratio, the characteristic of thinness, and the specific surface (expressed in m$^2$/100 g strands). The characteristic of thinness of the strands was determined with a new relation (Equation (2)), simpler and with a better accuracy than that given by Equation (1).

$$k_z \;=\; l \div \sqrt{t^2 + w^2}$$

(2)

where $d_e$—the equivalent diameter of the cross-section of the strand, in mm; $l$—the length of the strand, in mm; $t$—strand thickness, in mm; $w$—strand width, in mm.

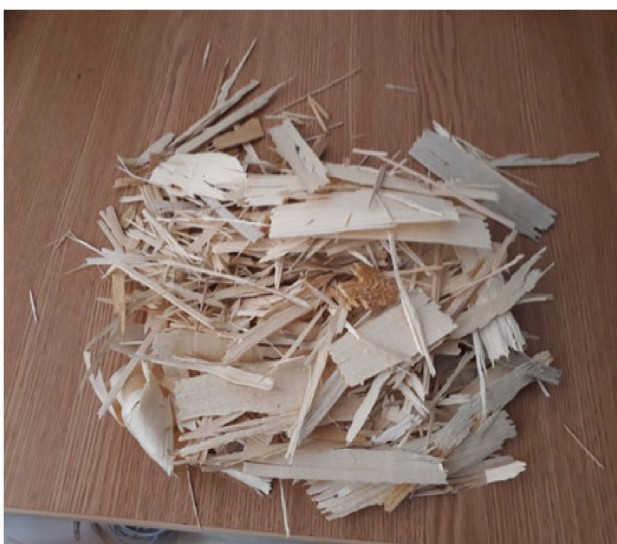

**Figure 5.** Strand size for OSB boards.

From the wood of the analysed species, 10 specimens with dimensions of $20 \times 20 \times 30$ mm were taken to determine the density of the wood species. The total quantity of strands from the four wood species was used to obtain four types of OSB boards in laboratory conditions, with the flat dimensions of $1200 \times 600$ mm, the density of 700 kg/m$^3$ and the use of 6% isocyanine adhesive type LUPRANATE M20S, one solvent-free adhesive based on diphenyl methane diisocyanate (MDI) produced by BASF Company, Ltd., (Jung-gu, Seoul, Korea).

The proportion of the front and core layers in the mat composition was: 1:2:1. The parameters of the pressing process were: specific pressure 3 MPa, temperature 160 °C, duration of 10 min for a plate thickness of 12 mm and a gradual discharge of the press in three stages. After pressing, the OSB boards were conditioned, hemmed edges and cut into minimal 30 specimens for density (EN 323), MOR, MOE, and Internal bond determinations, according to European standards [34–38]. The dimensions were determined on the test pieces with an electronically calliper and the resistances were tested on a universal testing machine type IB 600 (IMAL, San Damaso, Italy).

The obtained results for strands and OSB boards were statistically processed using Minitab 18 software (including Analysis of Variance ANOVA and Tukey Pairwise Comparisons), with a 95% confidence interval and Microsoft Excel, to highlight the differences between species and the influence of the strands characteristics on the properties of OSB boards.

## 3. Results and Discussion

The obtained results were referred to the characteristics of the strands, the characteristics of the OSB boards and the correlations between them, depending on the four wood species used (poplar, willow, pine, and spruce).

For the characteristics of the strands, all the dimensional characteristics of the measured strands made with the electronic callipers were put in the Excel tables, and then the global characteristics were calculated. They were recorded starting with the dimensions (l, w, and t) and ending with the characteristic of thinness.

Regarding the slenderness ratio, a high variability from 75.1 to 250.2 were found, while the thinness characteristic has lower values (2.3–6.86) and it is more comprehensive than the slenderness ratio. From the analysis of the table values, it is observed that the surface of the strands is very different from one strand to another, with differences of over 196%. For surface calculating of each strand, the surface of the edges was neglected, that was in fact very small. For example, in the case of poplar strand, the surface of the edges was 3.04 cm$^2$, which means less than 5% of the total surface of the strand, and for willow

strands, the ratio is even smaller due to the smaller thickness of the strand. The data for all the sample of strands were summarized in Table 1.

**Table 1.** Systematization of data regarding the average characteristics of the strands.

| Wood Species | *l* (mm) | *w* (mm) | *t* (mm) | *s* (m²/100 g) | *λ* | *D* (mm) | *K* |
|---|---|---|---|---|---|---|---|
| Poplar | 86.79 (23.2–119.5) * | 11.22 (2.46–43.7) | 0.93 (0.28–2.45) | 6214.1 | 121.0 | 11.2 | 10.5 |
| Willow | 88.45 (10.6–122.9) | 16.91 (3.89–49.7.) | 0.91 (0.22–2.03) | 7840.8 | 98.1 | 16.9 | 12.6 |
| Spruce | 80.55 (21.4–121.01) | 10.67 (2.3–40.75) | 1.06 (0.18–2.26) | 6071.5 | 86.6 | 10.7 | 10.2 |
| Pine | 77.10 (13.2–124.4) | 11.17 (1.2–52.1) | 0.92 (0.23–1.19) | 6347.8 | 92.6 | 11.6 | 10.4 |

* The values in parentheses are the minimum and maximum limits of the confidence interval.

As shown in Table 1, on the whole, the characteristics of the strands are different and from these values it is very difficult to draw a conclusion. However, it is clearly observed that willow strands have supremacy in terms of the length, width, and thickness of the strand; the specific surface; and the characteristic of thinness.

To perform a hierarchy of the quality of the woody species, a sorting algorithm is considered applying a maximum score of four points for the woody species that holds the first place, three points for the second place, two points for the third place, and one point for the last place. It is also considered that there are six criteria, eliminating the equivalent diameter which exclusively depends on the width and thickness of the strands. Summing the obtained score by each species strands, it is obtained the hierarchy from Figure 6.

Because there is a great variability of the surface of the strands (Table 1) and the average of these dimensions does not offer a clear separation of the species, but mainly because the dimensions of the strands depend on the cutting type, the settings of the machine and the degree of cutting wear, it was necessary to create a theoretical model to obtain the specific surface of the strands. The modelling of the specific surface of the strands started from the cut species and its average thickness. For this, it started from 100 g of strands, equivalent to 100 g of wood of each analysed species (evidenced by the density at 10% moisture content of each species).

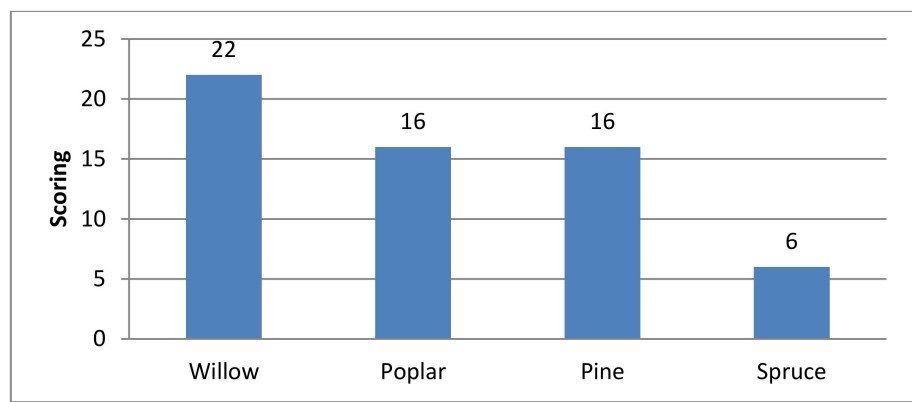

**Figure 6.** Scoring of the four analyzed strands.

The density of the wood of the species used was: spruce—405 kg/m³, pine—443 kg/m³, poplar—452 kg/m³, and willow—478 kg/m³. It was then considered that each wood piece of 10 × 10 mm was sliced with thicknesses from 0.1 mm to 2 mm, obtaining each time a new

strand surface (because only the surface of the strand face was considered active for gluing). Based on this algorithm, we found a calculus relation of the specific surface (Equation (3)).

$$S_s = 0.2 \cdot (\rho_{10} \cdot t)^{-1} \ [\mathrm{m^2/100\,g}] \tag{3}$$

where $\rho_{10}$ is the density at 10% of the woody species, in g/cm$^3$; $t$-strand thickness, in mm.

Using this Equation (3), the graph from Figure 7 was obtained. Even though the four regression curves are very close (because wood densities are appropriated), the power regression coefficient types are different for the analysed species. It can be observed that we have very good surfaces of over 1 m$^2$/100 g up to a thickness of 0.5 mm, after which the influence of the thickness on the specific surface is weak and almost constant.

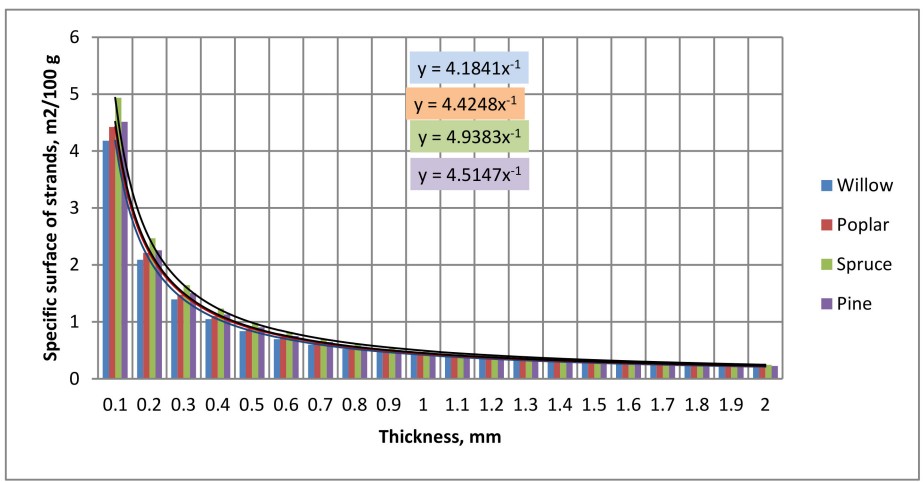

**Figure 7.** Modeling the specific surface of the strands according to species and thickness.

### 3.1. Characteristics of OSB Boards

Density, modulus of rupture and elasticity (MOR and MOE) and resistance to internal bond (IB) were analyzed separately.

### 3.1.1. OSB Board Density

It is observed (Figures 8 and 9) that the density of the OSB boards was about 30% higher than that of the solid wood from which it came, this being determined by the adhesive in the board and the exerted pressure (about 3 MPa) which has compressed and compacted the strand mat. Also, the density of the two species of fast-growing hardwoods was about 12% higher than that of the softwood ones, a fact also observed by other authors [26].

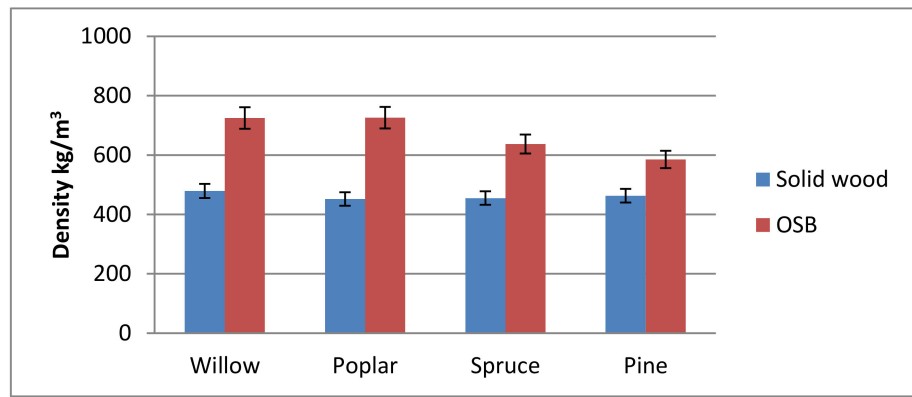

**Figure 8.** Density of OSB boards compared to solid wood.

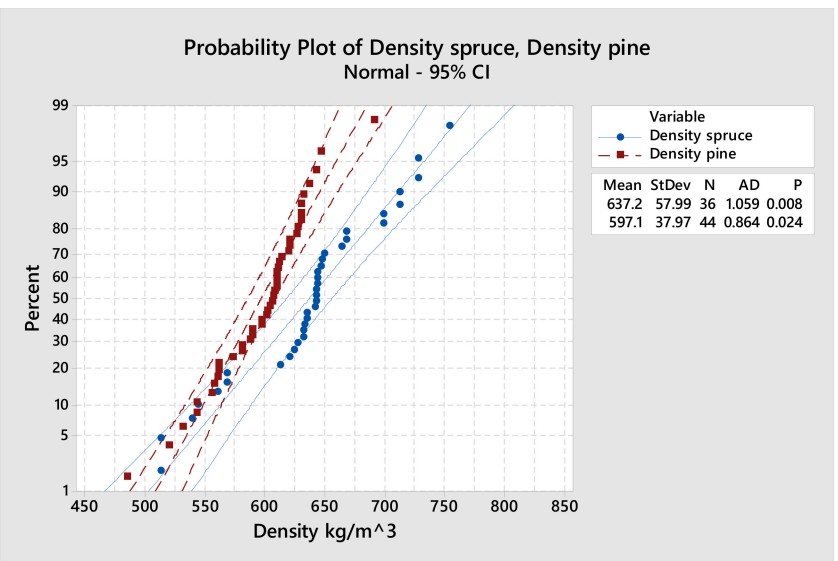

**Figure 9.** Probability plot for spruce and pine OSB density.

Analysis of variance (ANOVA) with null hypothesis and significance level of 0.05 was used and was obtained a good significance F-value of 4.57%. In this way the range of values was determined for 95% confidence interval as was for spruce species (621.65–652.65). Also, by the same method it was determined Tukey Pairwise Comparisons with the obtained order: poplar, willow, spruce, and pine.

### 3.1.2. MOR Results

MOR had the highest value of 44.54 N/mm$^2$ in the direction of board length (major axis) and 36.09 N/mm$^2$ in the direction of board width (minor axis) for willow and the smallest for spruce with 27.07 N/mm$^2$ on major axis and 20.65 N/mm$^2$ on the minor axis. It is observed (Figure 10) small differences between the values along the board compared to the side of the board, respectively of maximum 33.9% for poplar and minimum 20.3% for pine, which explains the stratification of the faces and the core and the ratio between layers of 1:2:1. Also, the values comply with the minimum conditions of standards [1] and [39] and agree with the values found by other authors for pine and poplar [26].

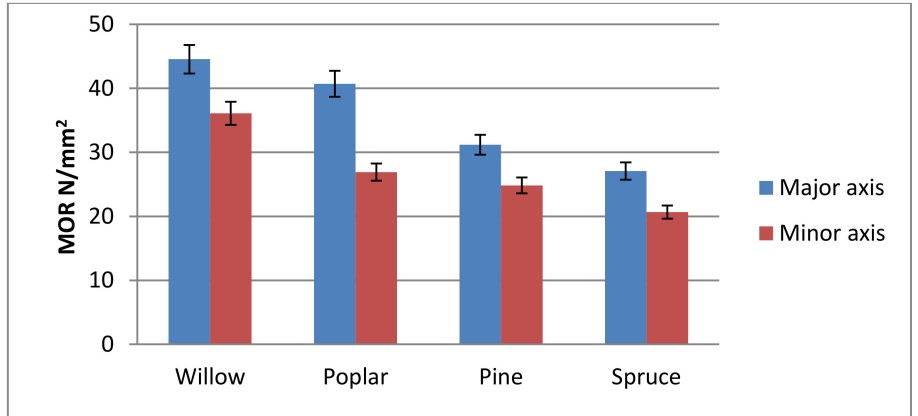

**Figure 10.** Modulus of rupture (MOR) for OSB boards of willow, poplar, pine, and spruce on major and minor axis.

### 3.1.3. MOE Results

As the modulus of rupture was, MOE was higher for hardwood species (the highest value recorded for willow trees with 4732 N/mm$^2$ in the major axis direction) and lower

for resinous (the smallest value being for pine 3020 N/mm$^2$ in the direction of board width) (Figure 11).

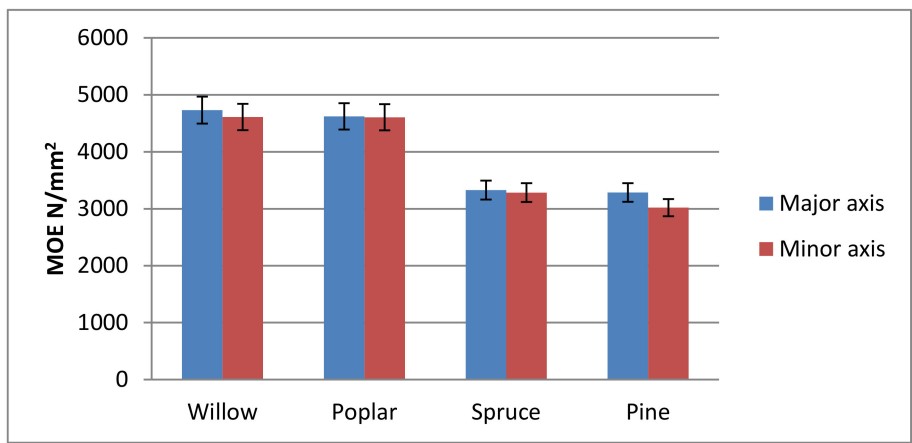

**Figure 11.** Modulus of elasticity (MOE) for the willow, poplar, spruce, and pine OSB boards on major and minor axis.

### 3.1.4. IB Results

The internal bond of the OSB boards was over 0.87 N/mm$^2$ for the dry boards and over 0.6 N/mm$^2$ for the boiled boards in water for two hours (Figure 12). It is observed that the fast-growing boards (willow and poplar) have values up to 34% higher than the two species of resinous (pine and spruce). The obtained values correspond to the values described by other authors [6], but also with those imposed by the companies that produce their own products [32].

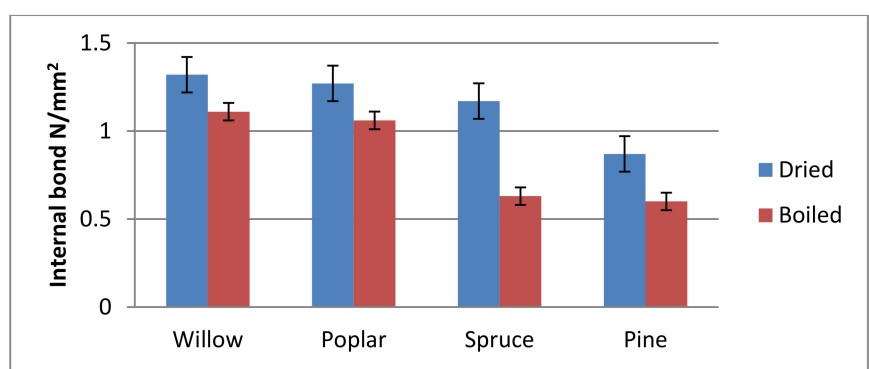

**Figure 12.** Internal bond for OSB dried and boiled boards.

Table 2 shows a centralization of all the characteristics of the OSB boards. At first glance, it is not possible to see very clearly which of the four types of boards are better or worse, for which the same hierarchical algorithm was applied as for the strands (Figure 7).

**Table 2.** OSB boards characteristics.

| Specie | Density, kg/m$^3$ | MOR 1 N/mm$^2$ | MOE 1 N/mm$^2$ | MOR 2 N/mm$^2$ | MOE 2 N/mm$^2$ | IB 1 N/mm$^2$ | IB 2 N/mm$^2$ |
|---|---|---|---|---|---|---|---|
| Poplar | 726.7 | 40.7 | 6623 | 26.9 | 2807 | 1.2 | 1.0 |
| Willow | 725.9 | 44.5 | 4732 | 36.0 | 4611 | 1.3 | 1.1 |
| Spruce | 637.2 | 20.5 | 3404 | 32.0 | 3913 | 1.1 | 0.6 |
| Pine | 585.7 | 30.6 | 3886 | 24.7 | 3020 | 0.8 | 0.6 |

Based on the above used algorithm, a number of different points were found from one board to another and the hierarchy was obtained, taking into account the seven characteristics of the OSB boards, respectively: density, MOR on major axis (MOR 1), MOE on major axis (MOE 1), MOR on minor axis (MOR 2), MOE on minor axis (MOE 2), IB on dried sample (IB 1), and IB on boiled samples (IB 2). In addition to the hierarchy of OSB boards chip characteristics were also tested, which can be seen in Figure 13.

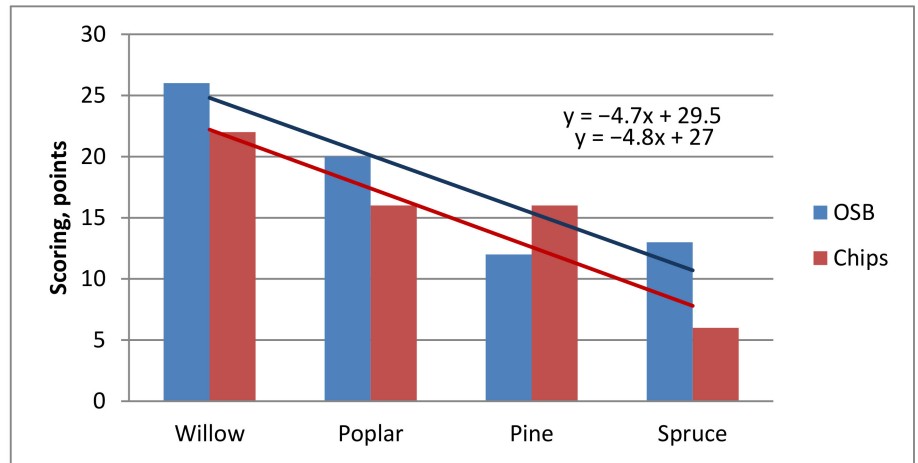

**Figure 13.** Hierarchy of wood species according to the characteristics of the chips and boards.

It is observed that there is an almost perfect parallelism between the linear regression curves (Figure 13). The two coefficients of the linear equations $m_1 = 4.7 = tg\alpha_1$ and $m_2 = 4.8 = tg\alpha_2$ offer two angles of 77.99 degrees and 78.23 degrees, respectively a difference of only 0.24 degrees.

The general conclusion of the above analysis is that there is an almost perfect correlation between the characteristics of the chips and the characteristics of the corresponding OSB boards. Also, from the entire study carried out in the paper, it is concluded that the species of fast-growing hardwood offers OSB boards with characteristics much better than the classic species of softwoods, and especially the pine. Such a new approach is needed for wood species used in the manufacture of OSB boards, by increasing the raw material base of this technology, respectively by adding new wood species (poplar and willow) cheaper and with superior properties in obtaining OSB boards. In fact, other authors have found that from the species of hardwood trees, OSB boards are produced and have some mechanical properties superior to those obtained from pine [4,26].

## 4. Conclusions

The paper evaluated the influence of the characteristics of the chips on the properties of the OSB boards obtained from the four species (poplar, willow, spruce, and pine) and found that this influence is overwhelming and positive.

The three-dimensional characteristics of the chips (length, width, and thickness) and three other synthetic qualitative characteristics (the slenderness ratio, the specific surface of the chips and the characteristic of thinness) were studied. Also, some methods, calculation relationships and algorithms for predicting these characteristics are emphases. The characteristics of the chips highlighted the superiority of the willow and poplar species over those of the resinous ones (spruce and pine).

The main physical–mechanical characteristics of OSB boards were analysed (density, modulus of rupture, and modulus of elasticity for bending strength on major and minor axis, and internal bond for dried and boiled specimens) from polar, willow, pine, and spruce chips. Also, it was highlighted that all the boards exceeded the minimum values required by the standards in the field. The superiority of the willow and poplar boards compared to the other analysed resinous species was highlighted with very good values of 44.5 MPa for MOR and 1.3. MPa for IB.

The comparative analysis of the corresponding OSB boards and their strands showed a very good correlation between them, finding that a good quality of the chips (given by size, the slenderness ratio, specific surface of the chips, and characteristic of thinness) will provide a good quality of OSB boards. In the same sense, it was observed that the best boards were those made from willow and poplar strands, due first and foremost to their very good quality. In this way, it has been demonstrated once again that the raw material basis of OSB technology is not static and is constantly improving.

**Author Contributions:** Conceptualization, A.L.; Methodology, A.L.; Software, A.L. and A.-E.D.; Validation A.L., A.-E.D., C.S., and V.-D.C.; Formal analysis, A.-E.D.; Investigation, A.L.; Resources, V.-D.C.; Data curation, A.-E.D., Writing—original draft preparation, C.S.; Writing—review and editing, A.L.; Visualization, A.L., Supervision, A.L.; Project administration, V.-D.C.; Funding acquisition, A.L. All authors have read and agreed to the published version of the manuscript.

**Funding:** This research received no external funding.

**Institutional Review Board Statement:** Not applicable.

**Informed Consent Statement:** Not applicable.

**Data Availability Statement:** Not applicable.

**Conflicts of Interest:** The authors declare no conflict of interest.

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
