# Peer review of "Influence of the Strand Characteristics on the Properties of Oriented Strand Boards Obtained from Resinous and Broad-Leaved Fast-Growing Species"

_applsci, doi:10.3390/app11041784_

Round 1
Reviewer 1 Report
The authors focus on an interesting problematics, namely "influence of the strand characteristics on the properties of OSB". Personally, I think that the results are predictable, but even such research is needed. The obtained results are in any case beneficial from the point of view of background for further research and possible application for potential producers of such boards.
I would like to propose few corrections and additions.
Formal shortcomings:
- DOI for articles in the references is missing,
- explanatory notes are incorrectly described for Figures 10 and 11,
- inappropriately inserted formulas,
- unexplained abbreviations, i.e. after each abbreviation used for the first time the full name should come,
- for all used equipment, products, etc., not only the country but also the city and manufacturer must be listed,
- inappropriate name for representative of woody species with resin.
Missing requisites:
- number of samples for each test (to complete surely),
- the statistical significance of the obtained results is related to the number of samples;but it’s all right, this is a requisite that is not absolutely necessary; but, respectively, for sets of less than 30 test samples, it is used to verify the statistical significance of the obtained results through the quantile of Student's distribution (e.g. level 0.95), the associated variability of measurement (COV) and the selected accuracy of measurement (e.g. 5%).
I consider the used research methods, as well as the introduction to the problematics, to be appropriate and sufficient. But for the assessment of the results, I would use a multi-factor analysis of variance ANOVA to evaluate the statistical significance of the differences, which SW Minitab 18 can definitely do.
However, the reproducibility of the results is difficult to predict, generally due to the high variability of wood properties. However, I believe that the trends are all right.
Author Response
For Reviewer 1
The authors would like to thank the reviewer for the time given to reviewing the paper, for the relevant remarks and for the improvements made to the work.
Formal shortcomings:
- DOI for articles in the references is missing,
- explanatory notes are incorrectly described for Figures 10 and 11,
- inappropriately inserted formulas,
- unexplained abbreviations, i.e. after each abbreviation used for the first time the full name should come,
- for all used equipment, products, etc., not only the country but also the city and manufacturer must be listed,
- inappropriate name for representative of woody species with resin.
Missing requisites:
- number of samples for each test (to complete surely),
- the statistical significance of the obtained results is related to the number of samples; but it’s all right, this is a requisite that is not absolutely necessary; but, respectively, for sets of less than 30 test samples, it is used to verify the statistical significance of the obtained results through the quantile of Student's distribution (e.g. level 0.95), the associated variability of measurement (COV) and the selected accuracy of measurement (e.g. 5%).
- I consider the used research methods, as well as the introduction to the problematics, to be appropriate and sufficient. But for the assessment of the results, I would use a multi-factor analysis of variance ANOVA to evaluate the statistical significance of the differences, which SW Minitab 18 can definitely do.
Author answers:
Formal shortcomings:
- The addition of DOI to each bibliographic reference is not mandatory in the Applied Sciences journal. See other published articles or manuscript preparation: https://www.mdpi.com/journal/applsci/instructions#preparation
- Figures 10 and 11 legends were completed. See lines 347 and 356.
- Equation 1 and 2 were changed. See lines 184 and 220.
- Some abbreviation as MDF (Medium-Fibre Density) and UNECE (United Nations Economic Commission for Europe) were explained. See lines 47 and 45.
- Data for all used equipment and products were added: See lines 232 and 243-236.
- Name for representative of woody species with resin was changed in “softwood”. See lines 320, 189 and 18.
Missing requisites:
- Number of samples for tests were added: See lines 232 and 240.
- The number of specimens for each test is determined by the standardized methodology. For example, in the case of resistance to static bending, 6 specimens on the major axis and another 6 specimens on the minor axis are required. At least 30 specimens were used in the paper (line 237, in order to obtain conclusive data. For example, in the case of density, 36 specimens were used for spruce and 44 specimens for pine, as shown in Fig. 9. However, the paper was completed with the analysis of ANOVA and its elements of statistical processing as seen in the 242-243- and 330-334-lines area. Tukey test was also used.
- The paper was completed with the analysis of ANOVA and its elements of statistical processing as seen in the 242-243- and 330-334-lines area.
Authors
Reviewer 2 Report
This is an interesting publication, and I would have accepted as is but for just some minor revisions. These are two:
- Please specify in the Experimental pressing time, pressing temperature, panels thickness and pressing cycle used for yiour experimental OSB.
- I think that what you call biphenyl methane bi-isocyanine is something never heard of, or is badly expressed in english. Isn't it diphenyl methane diisocyanate (MDI) you are talking about? If you do please correct throughout. If it is not than please add a structural chemical formula to explain what it is.
Author Response
Reviewer 2
The authors would like to thank the reviewer for the time given to reviewing our work, for the relevant remarks and for the improvements made to our work.
- Please specify in the Experimental pressing time, pressing temperature, panels thickness and pressing cycle used for your experimental OSB.
- I think that what you call biphenyl methane bi-isocyanine is something never heard of, or is badly expressed in English. Isn't it diphenyl methane diisocyanate (MDI) you are talking about? If you do please correct throughout. If it is not than please add a structural chemical formula to explain what it is.
Author answers
- Lines 233-236: The following completion was performed: The parameters of the pressing process were: specific pressure 3 MPa, temperature 160 0C, duration of 10 minutes for a plate thickness of 12 mm and a gradual discharge of the press in 3 stages.
- The name of the adhesive used in the experiments was corrected at line 231, respectively "diphenyl methane diisocyanate (MDI)"
Authors
Reviewer 3 Report
please see my comments attached

Author Response
For Reviewer 3
The authors would like to thank the reviewer for the time given to reviewing our work, for the relevant remarks and for the improvements made to our work.
- Reviewer: Line 14: examine or investigate. Word “establish” was replaced by “investigate”
- Reviewer: Line 15-20: Has to be re-written. Authors answer: These sentences were re-written.
- Reviewer: Line 41: please give a scientific reference for this. Authors: A reference was added, see line 42.
- Reviewer: Line 47: spelling. Authors answer: The abbreviation was opened: United Nations Economic Commission for Europe.
- Reviewer: Line 73: internal bond strength. Authors answer: The requested change (internal bond replaced the internal cohesion) has been made on lines 203 and 74.
- Reviewer: Line 99: other authors/other papers. Authors answer: The requested change has been made on lines 107, 121.
- Reviewer: Line 102: references are not relevant. Author answer: The presentation of these papers has been significantly reduced.
- Reviewer: Line 122: no relevant information. Authors answer: The presentation was reduced.
- Reviewer: Line 146: some comments as above. Authors answer: The presentation was reduced.
- Reviewer: Line 184: Fig 3/Eq. 2. Authors answer: Both the figure and the equation are related to the reference [33], so it is not the original of the paper, and only in the introductory part does it find its correct place. However, some changes have been made.
- Reviewer: Line 204: Objectives/Conclusion. Authors answer: The word “Objectives” has been deleted. A conclusion on the bibliographic study was introduced. Other small changes have been made to better understand the content of this paragraph. See lines 194-197.
- Reviewer: Line 236: parameters of pressing. Authors answer: The 3 parameters of press machine were added. See lines 233-236.
- Reviewer: Line 272: interesting approach. Authors answer: Thank you very much for appreciation.
- Reviewer: Line 320: differences in densities. Authors answer: There are many causes, but the most important is the density of the species from which the strands come, the density of softwood OSB boards will always be lower.
- Reviewer: Line 339: ANOVA. Authors answer: The paper was completed with the analysis of variance ANOVA and its elements of statistical processing as seen in the 242-243- and 330-334-lines area. Tukey test was also used.
Authors
Round 2
Reviewer 3 Report
The paper was greatly improved after the revision, my first round comments have been successfully addressed, therefore I am happy to suggest acceptance of the paper in its revised form